# European Green Deal: Study of the Combined Agricultural Aggregate

Volodymyr Nadykto [1,*], Volodymyr Kyurchev [1], Pavol Findura [2], Taras Hutsol [3,*], Sławomir Kurpaska [4], Anna Krakowiak-Bal [4] and Vyacheslav Vasyuk [5]

1 Department of Machine Operation and Technical Service, Dmytro Motornyi Tavria State Agrotechnological University, 18 B. Khmelnytskyi Ave., 72-310 Melitopol, Ukraine; volodymyr.kyurchev@tsatu.edu.ua

2 Department of Machines and Production Biosystems, Slovak University of Agriculture in Nitra, Trieda Andreja Hlinku 2, 949 76 Nitra, Slovakia; pavol.findura@uniag.sk

3 Department of Mechanics and Agroecosystems Engineering, Polissia National University, Staryi Blvd 7, 10-008 Zhytomyr, Ukraine

4 Department of Bioprocess Engineering, Power Engineering and Automation, University of Agriculture in Krakow, Mickiewicza Av. 21, 31-120 Kraków, Poland; s.kurpaska@urk.edu.pl (S.K.); anna.krakowiak-bal@urk.edu.pl (A.K.-B.)

5 Department of Electrical Engineering, Electromechanics and Electrotechnology, National University of Life and Environmental Science of Ukraine, 03-041 Kyiv, Ukraine; vasyuk@nubip.edu.ua

* Correspondence: volodymyr.nadykto@tsatu.edu.ua (V.N.); wte.inter@gmail.com (T.H.)

**Abstract:** The modern world industry involves the use of innovative approaches and optimisations of the existing agricultural management methods, which contribute to the implementation of the sustainable development of related industries and economies of different countries. The use of mobile agricultural units with extended functional properties can have a steady demand in the agricultural machinery market and contribute to the practical implementation of the philosophy of the "European Green Deal". The research results show that when assembling a unit for mowing agricultural crops with simultaneous grinding and placing their stubble in the soil, preference should be given to a self-propelled machine with rear swivel wheels. When using a wheeled tractor, it must have a reversible control post and a reversible transmission. A mathematical model of the collecting unit was developed, which allows for obtaining the corresponding amplitude and phase frequency characteristics and, with their help, the stability of the horizontal movement was evaluated. According to the results of field studies, the dispersion of the angle of directional oscillation of the tractor with front-steered wheels was 4.48 grad$^2$. For the tractor with rear-steered wheels, the value of this statistical parameter was 2.90 grad$^2$, which, according to the F-test at the level of statistical significance of 0.05, is naturally lower.

**Keywords:** amplitude frequency characteristic (AFC); disc harrow; European Green Deal; header; phase frequency characteristic (PFC); sustainable

## 1. Introduction

Agriculture is one of the integral branches of the economy of any country in the world. Modern trends and sustainable development programs encourage innovations and optimisation of existing farming methods, especially in European countries and countries with significant agricultural potential, such as Ukraine [1,2].

Depending on the prevailing climatic conditions or other circumstances, cereal crops and grasses are most often harvested almost all over the world, either in a one-phase or two-phase method [3,4]. The latter is implemented in practice by trailed and mounted harvesting units. The classic trailed unit comprises a self-propelled vehicle and a header to the right.

One of the significant problems of using such units is their reduced running sustainability in the horizontal plane. Its source is the momentum created by the header traction

resistance. Concerning the longitudinal axis of the tractor's symmetry, this force acts on a shoulder that exceeds half the harvesting unit operating width. The momentum formed in this way tends to deflect the tractor to the right from the line of its movement direction. In response, the tractor driver is often forced to keep the tractor's steering wheels turned at a certain angle to the left.

In addition to the tractor, the trailed header is also characterised by reduced running sustainability. In the movement process, it tends to turn in a horizontal plane in a clockwise direction. As a result, there is a decrease in its operating width, the quality of the harvested agricultural crop cut worsens, and the curvilinearity of the swaths increases with a simultaneous reduction in their uniformity in thickness, etc. [5].

The following are known among the solutions aimed at eliminating the above disadvantages: The header having an additional spring-loaded cable that keeps it from turning clockwise. Other researchers have equipped the right support wheel of the trailed header with an active hydraulic drive [6,7]. The speed of its rotation is set such that the header deviation angle from the direction of movement is constantly in the zone close to zero.

Another effective solution is to initially set the left and right wheels of the header at some angle concerning the harvesting unit movement direction [8]. In this case, the lateral forces on the header wheels create an offsetting momentum trying to turn it counterclockwise. A certain problem with applying this solution is that the initial angle of the header wheels depends on soil moisture and its mechanical composition. But these parameters can be different even within the same field.

Ultimately, these proposals were not widely used due to the header design's complexity and corresponding price rise. A more straightforward solution to this problem is to use mounted harvesting units equipped with front headers. According to their layout diagram, these units are symmetrical and asymmetrical. In a symmetrical harvesting unit [9–12], the crop swath is located between the wheels of the left and right sides of the self-propelled vehicle.

Such harvesting units are widely used for mowing tall miscanthus [13] and laying down sorghum sudangrass [14]. In the latter case, the headers are equipped with rotary working bodies that can work more efficiently with the lodged stalks of the harvested crops. Known are symmetrical harvesting units that use dual-purpose headers [15]. These machines are capable of forming both side and centre windrows. The self-propelled vehicle, in this case, has a wide track. Recently, harvesting units have been increasingly used, in which three harvesters are hitched on a self-propelled vehicle (tractor) as follows: one front and two sides [16]. Such units are characterised by high performance and low specific fuel consumption; moreover, their use helps reduce the compacted area of the field.

If a swath is formed to the left or right of a front-mounted header, it is usually asymmetrically attached to the self-propelled vehicle [4,17,18]. However, its lateral displacement is so small that it usually does not cause problems with the sustainability of the harvesting unit's horizontal movement.

Note that when implementing both of the considered methods of harvesting agricultural crops and grasses, their stubble remains directly exposed to sunlight. To reduce moisture loss and control weeds, it (simultaneously with the soil) should be chopped by disc cultivators (harrows) to a depth of 6–8 cm no later than 2–3 days. Studies show that this timely technological operation ensures moisture preservation in the soil within 2–6 mm.

Moreover, soil moisture accumulation significantly increases when chopping the stubble of an agricultural crop without a break in time simultaneously with its mowing. In practice, attempts are known to implement this technological method with a unit as part of a combined header and a trailed cultivator [19]. But the latter's presence significantly complicates the movement of such a harvesting unit in reverse when manoeuvring on the headland. Since the motion dynamics of such a dynamic system remained unexplored, and its scheme and design parameters were not substantiated, it did not find practical application in the future.

Analysing the above material lets us assume that the front harvesting and rear tillage machines must be mounted. At the same time, if the first (header) must have only a fixed one, then the second (harrow, cultivator) can have both a fixed and a swivelled joining in a horizontal plane with a self-propelled device. The latter can be equipped with both front- and rear-steering wheels. In this case, the following aspect is essential. The technical side of solving the fixed or swivelled joining of the tillage machine to the tractor does not present any particular difficulties. At the same time, the issue of choosing a self-propelled vehicle with front- or rear-steering wheels is quite problematic and, therefore, should be solved first.

In practice, units for harvesting root crops with mounted front and rear machines [20,21] are known. At the same time, the front machine cuts the haulm, and the rear one digs out the root crops. But firstly, these are units of a completely different technological purpose, and their use is not intended to conserve soil moisture. Secondly, these publications do not consider the harvesting units' movement dynamics or their statics. Any publications with this or similar content have not been found.

At the same time, it should be remembered that the high-quality functioning of such a harvesting unit as a dynamic system is possible only if its movement in the horizontal plane is highly stable. Sustainability should be understood as a satisfactory response of a dynamic system to a disturbing action. The system we have proposed for consideration is a tracking system. The sustainability of its movement can be estimated using the corresponding amplitude (AFC) and phase (PFC) frequency characteristics. The harvesting unit whose real AFC and PFC are closest to perfect is the most preferred. As is known, when a disturbing action is processed, the perfect AFC equals 0, while the PFC tends to infinity [22].

It is pretty clear that an appropriate mathematical apparatus is needed for a theoretical study of motion dynamics with the subsequent justification of such a unit's diagram and design parameters. Since it is currently absent, this article aims to design a mathematical model that allows us to evaluate the sustainability of the unit's horizontal movement based on a self-propelled vehicle with front- and rear-steering wheels, joined with a front header and a tillage machine mounted at the back. These studies will ensure the financial security of agricultural enterprises [23] to develop the agricultural sector of Ukraine [24].

## 2. Theoretical Premises

According to the above aim, we will consider the harvesting unit movement dynamics as part of a self-propelled vehicle (tractor), a front-mounted header, and a mounted tillage machine. The front header in the horizontal plane is fixed to the tractor, thus forming a joint centre of mass located in point S and connected to the YOX movable coordinate system (Figure 1).

In the differential form of writing, the horizontal movement mathematical model of the considered harvesting units has the following view:

$$A_{11} \cdot \ddot{X}_s + A_{12} \cdot \dot{X}_s + A_{13} \cdot \dot{\varphi} + A_{14} \cdot \varphi + A_{15} \cdot \beta = f_{11} \cdot \alpha + f_{12};$$

$$A_{21} \cdot \ddot{\varphi} + A_{22} \cdot \dot{\varphi} + A_{23} \cdot \varphi + A_{24} \cdot \dot{X}_s + A_{25} \cdot \beta = f_{21} \cdot \alpha + f_{22}; \qquad (1)$$

$$A_{31} \cdot \ddot{\beta} + A_{32} \cdot \dot{\beta} + A_{33} \cdot \beta + A_{34} \cdot \dot{\varphi} + A_{35} \cdot \varphi + A_{36} \cdot \dot{X}_s = 0,$$

where

$$A_{11} = M_a;$$
$$A_{12} = (k_a + k_b + P_{fa} - F_b) \cdot V_0^{-1};$$
$$A_{13} = \pm [(k_a + P_{fa}) \cdot (L - b) + (F_b - k_b) \cdot b] V_0^{-1};$$
$$A_{14} = -A_{12} \cdot V_0;$$
$$A_{15} = P_r;$$
$$A_{21} = J_a;$$
$$A_{22} = \left[ (k_a + P_{fa}) \cdot (L - b)^2 + (k_b - F_b) \cdot b^2 \right] V_0^{-1};$$
$$A_{23} = -A_{13} \cdot V_0;$$
$$A_{24} = A_{13};$$
$$A_{25} = -P_r \cdot (L + l_z - b) \; - \; \text{Revers};$$
$$A_{25} = P_r \cdot (l_z + b) \; - \; \text{Forward};$$
$$A_{31} = J_m;$$
$$A_{32} = (k_k + P_{fk}) \cdot l_k^2 \cdot V_0^{-1};$$
$$A_{33} = (k_k + P_{fk}) \cdot l_k;$$
$$A_{34} = (k_k + P_{fk}) \cdot (L - b + l_z) \cdot l_k \cdot V_0^{-1};$$
$$A_{35} = A_{33};$$
$$A_{36} = -A_{33} \cdot V_0^{-1};$$
$$f_{11} = k_a;$$
$$f_{12} = R_p;$$
$$f_{21} = \pm (L - b) \cdot k_a;$$
$$f_{22} = R_p \cdot (b + l_c) - R_k \cdot \Delta \; - \; \text{Revers};$$
$$f_{22} = R_p \cdot (L - b + l_c) - R_k \cdot \Delta \; - \; \text{Forward}.$$

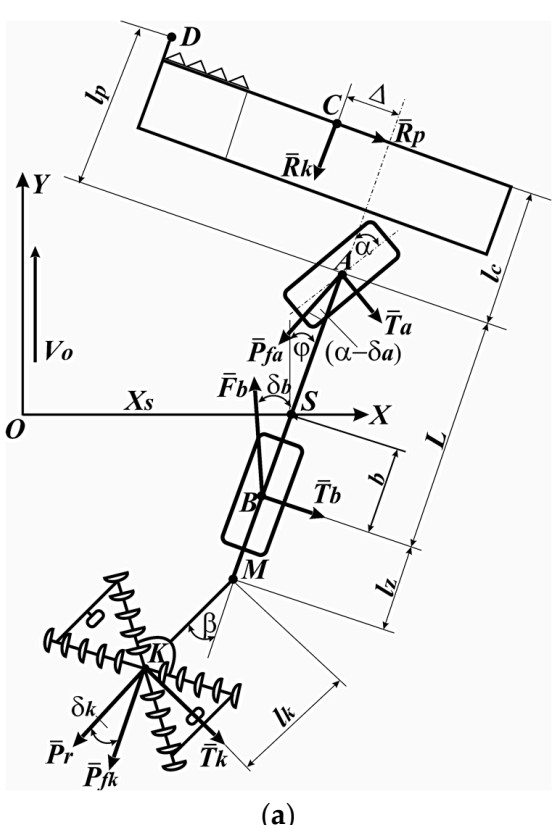

(**a**)

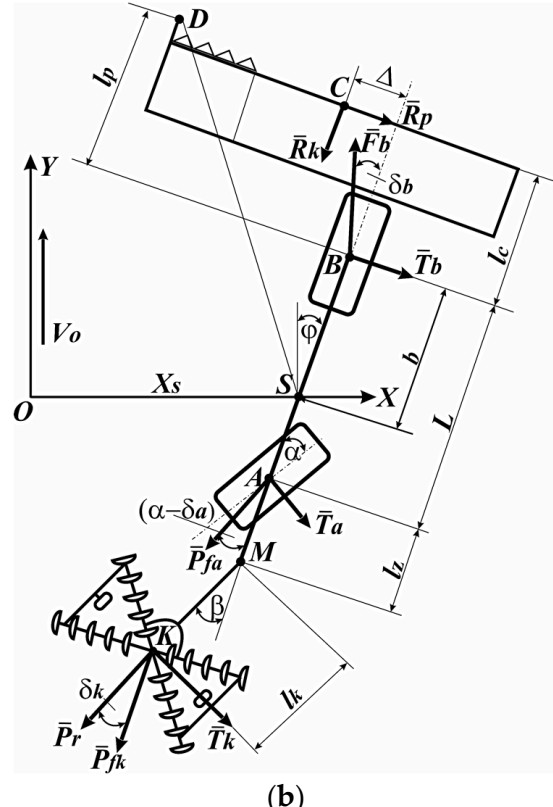

(**b**)

**Figure 1.** Harvesting unit: (**a**) with tractor's front-steering wheels; (**b**) with tractor's rear-steering wheels.

In the equations for the coefficients $A_{13}$ and $f_{21}$, the "+" sign corresponds to the forward movement of the tractor and "−" to the reverse one. The equations system (1) includes the following parameters: $M_a$ is the mass of the tractor with a mounted header, kg; $J_a$ is the tractor and header inertia moment relative to the vertical axis passing through point S (see Figure 1), kg·m$^2$; $J_m$ is the disk harrow inertia moment about the axis passing through point M, kg·m$^2$; $k_a$, $k_b$, $k_k$ are coefficients of resistance to tire yaw of the tractor front and rear wheels, as well as the wheels of the disc harrow, respectively, kN·rad$^{-1}$; L, b, $\Delta$, $l_c$, $l_z$, $l_k$ are linear design parameters of the unit (see Figure 1).

The input variables in the equations system (1) are as follows: (i) the control action in the form of the tractor's steered wheels rotation angle ($\alpha$); (ii) disturbing action from the force $R_p$; (iii) the total disturbing effect expressed by the factor $f_{22}$.

Based on the mathematical model (1), the amplitude (AFC) and phase (PFC) frequency characteristics of the disturbances processing by the dynamic system can be constructed. To design these AFCs and PFCs, an appropriate transfer function is needed. To assess the sustainability of the dynamic system motion, in this case, the one that reveals the mechanism of the impact of a disturbance on the nature of changes in the tractor heading angle is suitable. Consider the momentum (Mr) acting on the unit as such an external perturbation. The equation for its calculation is:

For tractor with front-steering wheels:

$$M_r = f_{22} = R_p \cdot (b + l_c) - R_k \cdot \Delta ; \tag{2}$$

For tractor with rear-steering wheels:

$$f_{22} = R_p \cdot (L - b + l_c) - R_k \cdot \Delta. \tag{3}$$

The transfer function [$W_M (p)$], in this case, will be:

$$W_M(p) = \frac{D_M}{D_S} = \frac{\begin{vmatrix} K_{11} & 0 & K_{13} \\ K_{21} & f_{22} & K_{23} \\ K_{31} & 0 & K_{33} \end{vmatrix}}{\begin{vmatrix} K_{11} & K_{12} & K_{13} \\ K_{21} & K_{22} & K_{23} \\ K_{31} & K_{32} & K_{33} \end{vmatrix}} \tag{4}$$

where

$$K_{11} = A_{11} \cdot p^2 + A_{12} \cdot p;$$
$$K_{12} = A_{13} \cdot p + A_{14};$$
$$K_{13} = A_{15};$$
$$K_{21} = A_{24} \cdot p;$$
$$K_{22} = A_{21} \cdot p^2 + A_{22} \cdot p + A_{23};$$
$$K_{23} = A_{25};$$
$$K_{31} = A_{36} \cdot p;$$
$$K_{32} = A_{134} \cdot p + A_{35};$$
$$K_{33} = A_{31} \cdot p^2 + A_{32} \cdot p + A_{33};$$
$$p = d/dt - \text{Laplace operator.}$$

In this transfer function: $D_{.M.}$ is a determinant that reflects the impact of the turning momentum $M_r$ on the tractor heading angle $\varphi$; $D_s$ is the main determinant of the equations system (1).

The equations system (1) is easy to convert into a mathematical model in which the tillage machine $\beta$ rotation angle can be considered a perturbing input action. In this case, the harvesting unit's movement sustainability can be estimated by AFCs and PFCs of

the dynamic system working out the perturbing action in the form of fluctuations in the parameter β. The transfer function required for this has the following format:

$$W_\beta(p) = \frac{D_\beta}{D_S} = \frac{\begin{vmatrix} K_{11} & A_{15} \\ K_{21} & A_{25} \end{vmatrix}}{\begin{vmatrix} K_{11} & K_{12} \\ K_{22} & K_{22} \end{vmatrix}} \tag{5}$$

where $D_\beta$—determinant reflecting the effect of the tillage machine rotation angle (β) on the tractor's heading angle φ.

### 3. Materials and Methods

When calculating the necessary transfer functions, as well as the corresponding AFCs and PFCs, the following values of the harvesting unit design parameters were used: $M_a$ = 9800 kg; $J_a$ = 53,550 kg·m$^2$; $J_m$ = 3800 kg·m$^2$; $k_a$ = 115 kN·rad$^{-1}$; $k_b$ = 80 kN·rad$^{-1}$; $P_{fa}$ = 5 kN; $F_b$ = 12.6 kN; $P_r$ = 7.5 kN; L = 2.86 m; b = 1.0 m; $l_z$ = 1.1–3.0 m; $l_c$ = 3.75 m; $\Delta$ = 0.75 m; $B_p$ = 6 m; $V_0$ = 2–4 m·s$^{-1}$.

To confirm the mathematical modelling results in the field, research was carried out on harvesting units as part of the HTZ-16131 tractor (Ukraine) with the ZHVN-6B header (Ukraine) and the BDN-3 disc harrow. In the first unit's variant, the tractor was set to forward motion (Figure 2), and in the second one, to reverse (Figure 3). To do this, its steering post was turned in the cab by 180 degrees, and the gearbox was switched to reverse mode. The disc harrow was hung on a tractor with the possibility of its rotation in the horizontal plane at an angle β = ±8°.

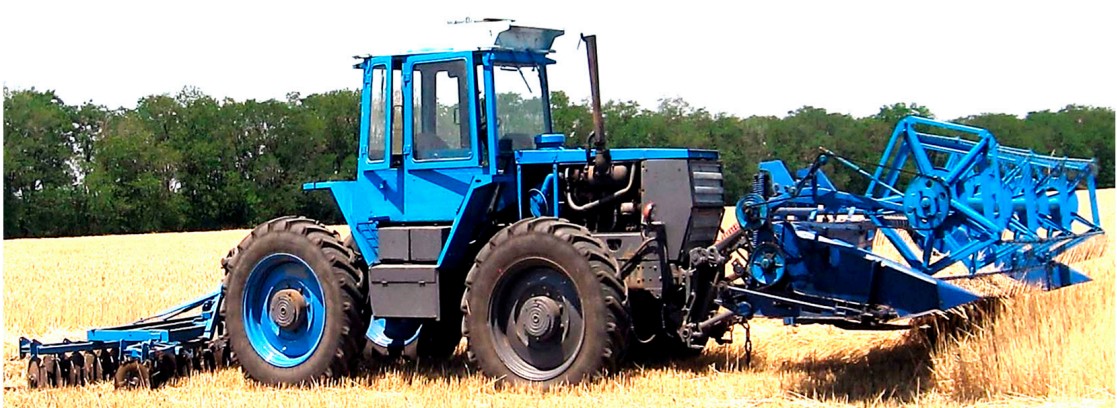

**Figure 2.** Harvesting unit with tractor's front-steering wheels.

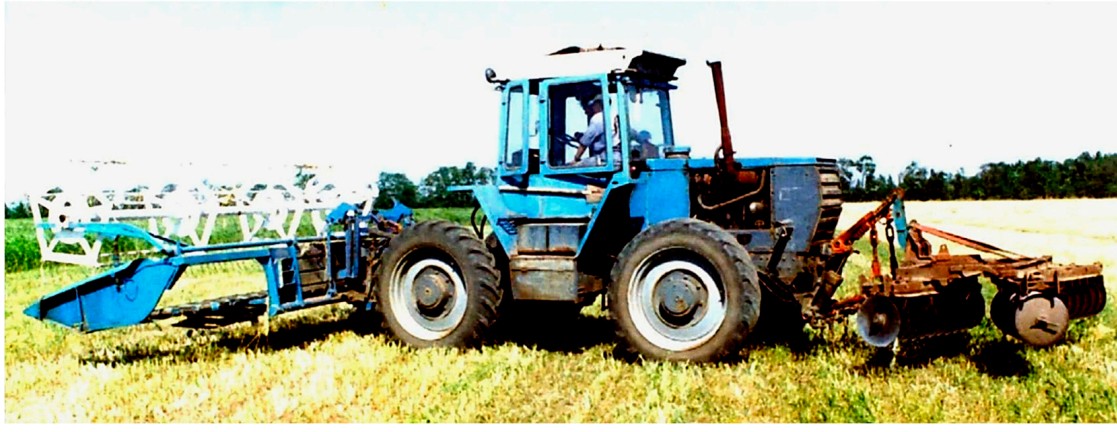

**Figure 3.** Harvesting unit with tractor's rear-steering wheels.

These harvesting units operated on mowing winter wheat. The experimental field was divided into sites 250 m long each. The first 50 m of each site was used to accelerate the harvesting unit. On the remaining 200 m, it performed a functional movement in three repetitions. The following formula calculated the velocity of its movement: $V_0 = 200 \cdot t^{-1}$ (m·s$^{-1}$). Here, t is the time for the harvesting unit to pass through the site. We used an electronic stopwatch KHP PC3860 (China), to register it, with a measurement error of 0.01 s.

The tractor's heading angle ($\varphi$) was recorded during the harvesting unit's functional movement using Arduino Uno (Italy). A Gy-521 MPU-6050 gyroscope (China) was used to measure this parameter, which was installed on the tractor's hood at the point of its mass centre.

The values of the angle $\varphi$ read by the Arduino Uno using the SD SPI module (China) were written to a CD card. The digitized experimental data were used to calculate variances and normalised spectral densities of the tractor heading angle $\varphi$ oscillations.

Before carrying out experimental studies of harvesting units, the winter wheat harvested area characteristics were determined. The following were measured: (i) soil moisture and density in the 0–10 cm layer; (ii) mean height of wheat plants, cm; (iii) wheat yield, ton·ha$^{-1}$; (iv) weeds' density, g·m$^{-2}$; (v) wheat stubble height, cm; (vi) disc harrow tillage depth, cm.

The instruments and methods for measuring soil moisture and density are substantively described by [25]. The height of plants and stubbles of winter wheat was determined along the field diagonal with a ruler 1 m long with a measurement error of $\pm0.5$ cm. The number of such measurements was 300, and the measurement step was 1 m.

The weeds' density in wheat crops and their yield were determined using a wooden frame with an area of 1 m$^2$. The distance between 30 measurement zones performed along the field diagonal was at least 10 m. The mass of weeds and wheat grains that fell into the frame zone after threshing its spikelets was determined using an Axis AD 200 electronic balance (Poland), the measurement error of which was 0.001 g.

The disc harrow tillage depth was measured along the site diagonal in two repetitions. The measurement step for this parameter was 0.2 m. The measurement number of this parameter in each repetition is 300. To measure the tillage depth, we used a kit created by us based on an ultrasonic sensor HC-SR04 (China) and an Arduino UNO board (Italy). The error in measuring the tillage depth by the kit does not exceed 0.5 cm.

### 4. Results

As follows from the analysis of the equations system (1), the external disturbance acting on the harvesting unit in the horizontal plane is the momentum described by Equations (2) or (3). As the frequency of its oscillations increases from 0 to 5 s$^{-1}$, the calculated AFCs (Figure 4) decrease, and the PFCs (Figure 5) increase. Note that both results are desirable. They are due to a stronger manifestation of the unit inertial properties in the role of a damper at higher frequencies of the momentum $M_r$ oscillations.

The calculation results show that the AFC of excitation processing by a harvesting unit based on a tractor with front-steering wheels (forward running) is worse than that of a unit based on a tractor configured for reverse movement. Moreover, the difference between these frequency characteristics (10–16%) is more noticeable at frequencies of the momentum $M_r$ oscillations close to zero (see Figure 5). To explain this result, consider Equations (2) and (3). The values of those parameters included in them are as follows: $R_p$ = 1500 N; b = 1.7 m; $l_c$ = 3.75 m; L = 2.86 m; $R_k$ = 7500 N; $\Delta$ = 0.75 m.

The calculations using Formulas (2) and (3) show that when using a tractor with front-steering wheels, the turning momentum acting on the harvesting unit in the horizontal plane is 2.55 kN·m. When using a tractor with rear-steering wheels as part of a harvesting unit, the value of this momentum is 1.74 kN·m, 31.8% less. For this reason, as follows from the analysis of Figure 4, the unit of such a scheme is less responsive to disturbing influences.

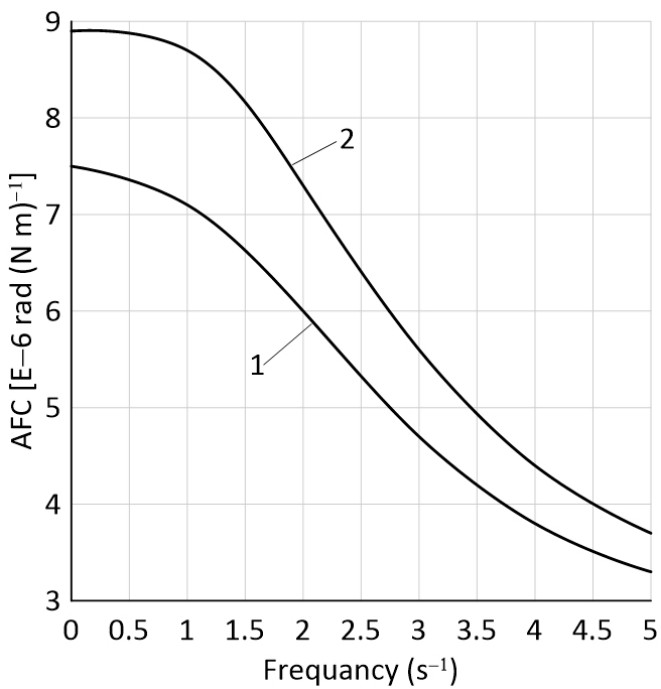

**Figure 4.** AFCs of working off by a harvesting unit momentum at reversing (1) and direct (2) movement of the tractor.

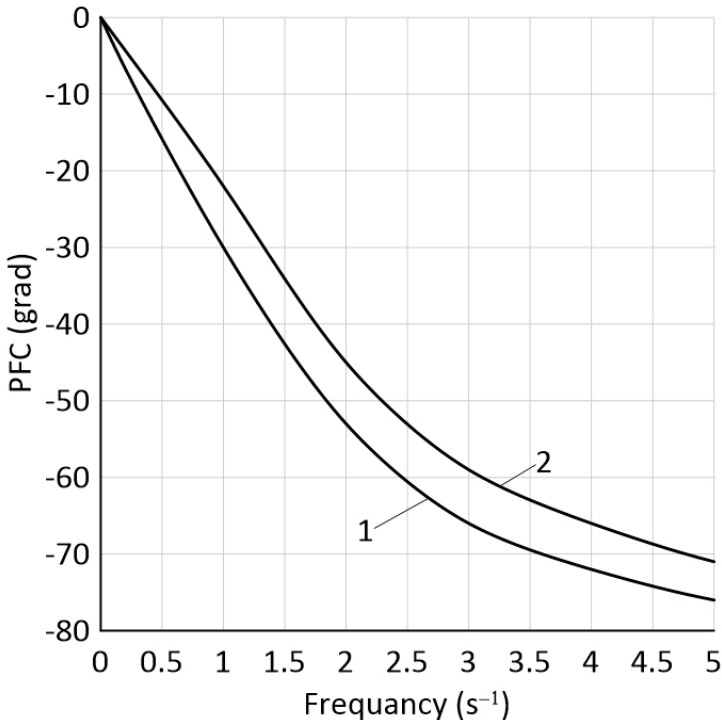

**Figure 5.** PFCs of work out by a harvesting unit momentum at direct (1) and reversing (2) movement of the tractor.

The delay in the response of the harvesting unit to the momentum $M_r$ action is, although not by much, in principle, greater than the tractor's forward motion (Figure 5). This result is generally quite adequate for the following reason. The middle of the tractor's front axle (p. A, Figure 1a) is located between the point of the disturbing forces $R_k$ and $R_p$ (p. C) application and the unit mass centre (p. S).

When the tractor runs straight, its front-steering wheels account for 62% of the total weight and only 38% for the rear wheels. From here, it is pretty logical to expect that the delay in the reaction of the tractor to the action of the moment $M_r$ will be large, with a greater vertical loading of its front axle than the rear one. This means that PFC will have large values, which is confirmed by the data in Figure 5 (curve 1).

Now let us estimate the harvesting unit movement sustainability when the deviation angle of the tillage machine ($\beta$) is not the output value of the dynamic system but the input value in the form of an external perturbation with transfer function (5). The results of mathematical modelling show that, in this case, the harvesting unit, based on a tractor with rear-steering wheels (reverse movement), is more stable. Its AFC is characterised by lower values (by 22%) (curve 2, Figure 6) than the similar amplitude-frequency characteristic of the unit based on a tractor with front-steering wheels (curve 1, Figure 6).

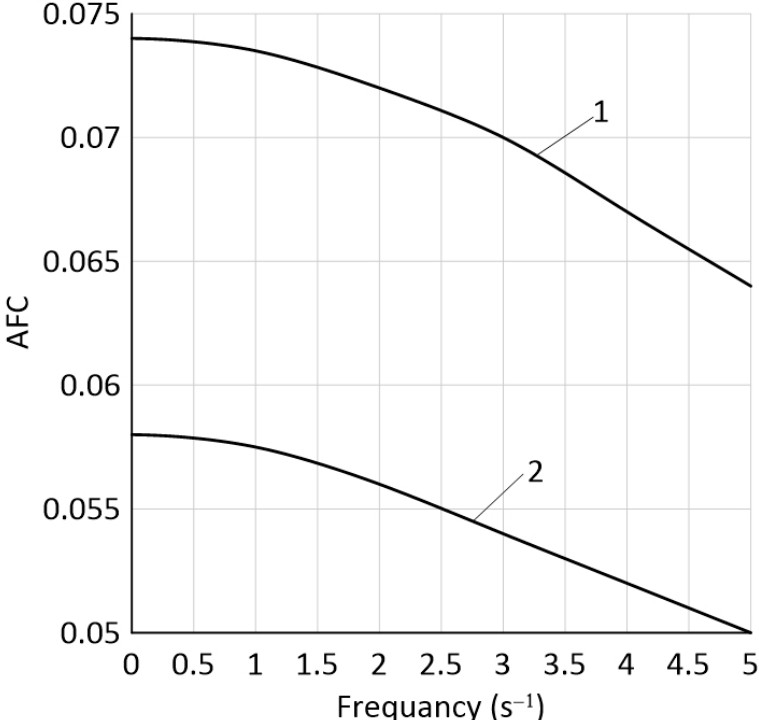

**Figure 6.** AFCs of harvesting units when they work out angle $\beta$ oscillations for forward (1) and reversing (2) tractor movement.

This result is because, compared to the tractor with front-steering wheels, the harvesting unit mass centre is closer to the point of application of the tillage machine's traction resistance transverse component $P_r$ (Figure 1b). This leads to a decrease in the shoulder, hence the momentum magnitude created by this force concerning the tractor.

It is quite natural that the specified momentum increases with an increase in the force $P_r$ transverse component value. This occurs either with an increase in the tillage machine deviation angle ($\beta$, Figure 1b) or with an increase in its traction resistance. In particular, this may be the case with an increase in the velocity of the operating movement $V_0$. Calculations show that a change in the $V_0$ parameter from 2 to 4 m·s$^{-1}$ causes a corresponding increase in the tractor heading angle. At the same time, the value and the parameter's $\varphi$ growth intensity are higher at a higher harvesting unit velocity (Figure 7).

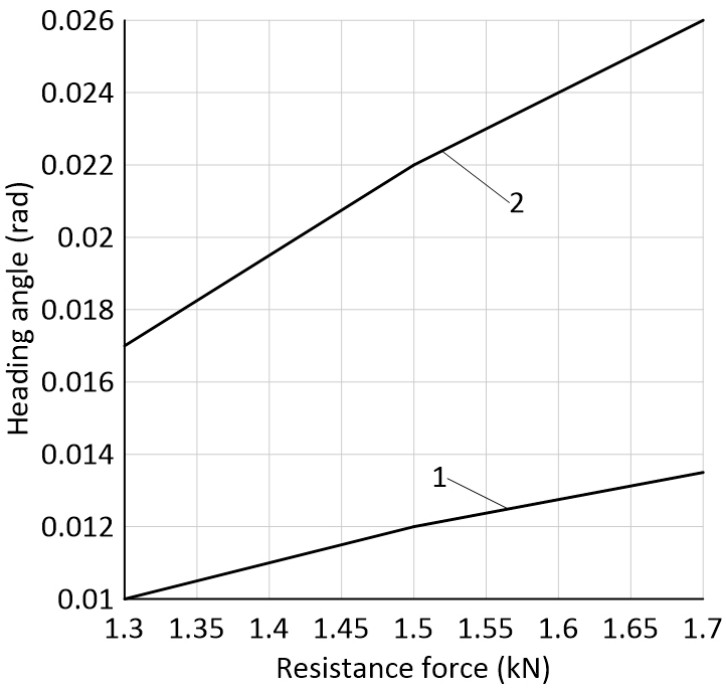

**Figure 7.** Dependence of the angle φ on the transverse component traction resistance tillage machines at different working velocities harvesting unit: 1—2 M·s$^{-1}$; 2—4 m·s$^{-1}$.

## 5. Discussion

Field research data fully confirm the above results of mathematical modelling. The conditions for conducting experimental studies of the harvesting units are presented in Table 1. The harvesting units operating velocity in both design diagrams was almost the same. Its mean value varied within $2.45 \pm 0.06$ m·s$^{-1}$.

**Table 1.** Characteristics of the winter wheat experimental site.

| Index | Value |
| --- | --- |
| Soil moisture in layer of 0–10 cm (%) | 14.9 |
| Soil bulk density in the layer of 0–10 cm (g·m$^{-3}$) | 1.29 |
| Winter wheat yield (ton·ha$^{-1}$) | 4.58 |
| Mean plant height (m) | 0.73 |
| Weeds density (g·m$^{-2}$) | 12.6 |
| Wheat stubble height (cm) | 13.2 |
| Stubble disc depth (cm) | 7.9 |

The data analysis shows that the tractor's heading angle oscillations variance with the rear-steering wheels was 2.90 grad$^2$. For a unit with front-steering tractor wheels, the value of this statistical index was higher and equal to 4.48 grad$^2$. The F-test calculated value for these variances is 1.54. This is less than its tabular value, which is 1.39 at the statistical significance level of 0.05. As a result, with a 95% confidence probability, it can be argued that the null hypothesis about the equality of the compared variances is rejected. And this means that the harvesting unit based on the tractor with rear-steering wheels is more stable. The heading angle oscillations variance of the latter is naturally smaller than the oscillations variance of the same parameter for a tractor with front-steering wheels.

It should be noted that using the rear wheels of a mobile vehicle as a steerable one is currently considered as a strategy for increasing the sustainability and manoeuvrability of the movement of both harvesting units and other means [26,27]. To study this issue, the

corresponding theoretical premises are being developed [28]. Research has established that using rear steerable wheels in a mobile vehicle can significantly improve the driver's working conditions. The researchers note that in this case, there is a decrease in the intensity of the control action by about 55% [29].

Let us add that a smaller variance and a narrower spectrum characterise the heading angle oscillations of the tractor with rear-steering wheels. Its cutoff frequency, as follows from the analysis of the normalised spectral density, does not exceed 2.2 s$^{-1}$ (curve 2, Figure 8).

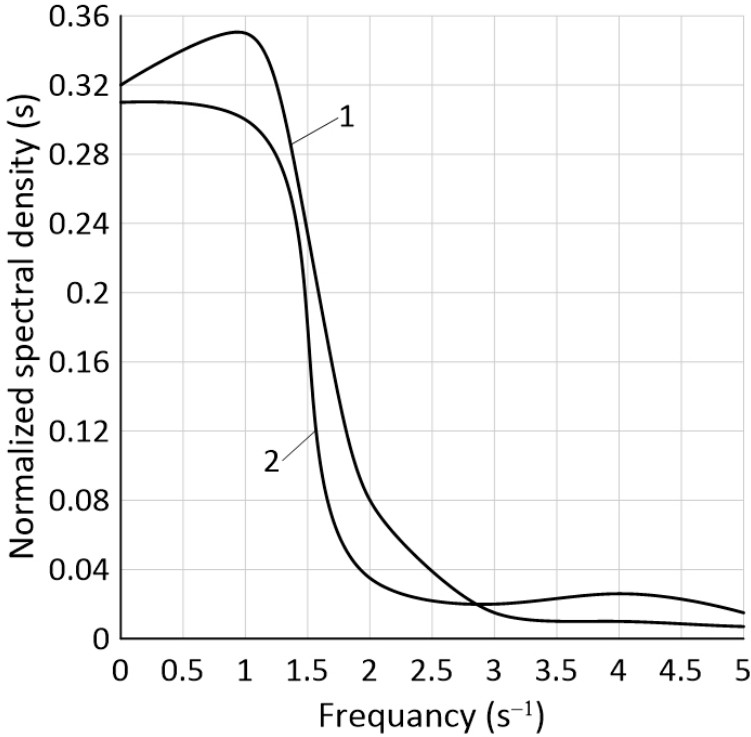

**Figure 8.** Normalised spectral oscillation densities of the tractor heading angle at its forward (1) and reversing (2) movement.

At the same time, the variance of heading angle oscillations of the tractor with front-steering wheels is concentrated in the range, the cutoff frequency of which is approximately 3.3 s$^{-1}$ (curve 1, Figure 8). All this indicates that the use of such a tractor as part of a harvesting unit with a front header and a rear-mounted tillage machine is more preferable.

It has already been emphasized earlier that studies of such a harvesting unit, which combines the mowing of grain crops with the chopping and incorporation of their stubble into the soil, have not been found. At the same time, an increase in the tractor heading angle oscillations amplitude with an increase in its velocity was recorded when harvesting sugar beet haulm with a trailed combine.

For mounted implements, such results were obtained in ploughing [30] and row crops [31]. At the same time, the machine-tractor units movement velocity mode was considered from 1 to 5 m·s$^{-1}$ [32].

Research has established that when moving modern agricultural units at a speed of up to 4.5 m·s$^{-1}$, a kinematic model can be used instead of their dynamic movement model. Its application allows us to obtain quite adequate results [33].

In confirmation of the legitimacy of the results obtained in this article, the fact of using rear-steered wheels in the design of field robots indicates. Studies show that the trajectory of their horizontal movement does not exceed ±10 cm in this case. The robot heading angle oscillations will not exceed ±2° [28].

In conclusion, we note that the practical application of the considered combined harvesting unit significantly expands the functionality of the tractor used in this case. Firstly, it allows the use of more agricultural machines aggregated with it. Secondly, such a tractor uses the front hitch linkage system more efficiently, which justifies its installation.

Thirdly, the sustainability movement of the unit based on it creates more comfortable conditions for the work of the tractor driver. Fourthly, using one harvesting unit instead of two (mower and tillage) reduces the pass number across the field. And this causes a decrease in the compacted area of the field. Together with a reduction in exhaust emissions, this contributes to the practical implementation of the European program "European Green Deal" [1,34]. In addition, as market research shows [35], a mobile device with such extended functional properties can be sustainable in demand in the agricultural machinery market.

## 6. Conclusions

The modern world industry involves the use of innovative approaches and optimisation of existing methods of agricultural management, which contributes to the implementation of sustainable development of related industries and economies of different countries.

According to the research results, it is possible to state the following:

1. The use of mobile agricultural units with enhanced functional properties can have a steady demand in the market of agricultural machinery and contribute to the practical implementation of the philosophy of the "European Green Deal";

2. A mathematical model has been developed that makes it possible to theoretically evaluate the stability of the plane-parallel movement in the horizontal plane of the harvesting unit for mowing agricultural crops simultaneously with their stubble grinding and planting in the soil. The unit includes a self-propelled machine (tractor) with front- or rear-steering wheels, a front header and a mounted tillage machine;

3. When using a tractor with rear-steering wheels, the harvesting unit is subjected to an impulse of 31.8% less than when using a tractor with front-steering wheels. This advantage is more noticeable at lower (and therefore more likely) frequencies of torque oscillations. As a result, the AFC value for working out the disturbing action of such a unit due to the stability of its movement is 10–16% less;

4. When working out the perturbation as the angle of rotation of the tillage machine, the harvesting unit based on the tractor with rear-steered wheels (reversible movement) is more stable. Its AFC is characterised by 22% lower values compared to the similar AFC of a unit based on a tractor with front-steered wheels;

5. According to the results of field studies, the dispersion of the angle of directional oscillation of the tractor with front-steered wheels was 4.48 grad$^2$. For a tractor with rear-steered wheels, the value of this indicator was 2.90 grad$^2$, which according to the F-test at the level of statistical significance of 0.05, is lower;

6. As a result of the analysis of theoretical and experimental research data, it is shown that when designing a unit for mowing agricultural crops with simultaneous cutting and planting of their stubble in the soil, preference should be given to a self-propelled machine with rear-wheel drive. If a wheeled tractor is used for this, it must have a reversible control post and a reversible transmission;

7. The practical application of the considered combined unit significantly expands the tractor's functionality. It allows the use of a larger number of agricultural machines aggregated with it, the front linkage system is used more efficiently, and more comfortable conditions are created for the tractor operator, the use of one harvesting unit instead of two (mower and tiller), which, in turn, it reduces the number of passes on the field and makes it possible to reduce soil compaction and exhaust gas emissions.

**Author Contributions:** Conceptualization, V.N. and T.H.; methodology, V.K. and S.K.; literature review, P.F.; software, V.N. and A.K.-B.; validation, V.V.; formal analysis, P.F.; supervision, T.H. All authors have read and agreed to the published version of the manuscript.

**Funding:** Financed from the subsidy of the Ministry of Education and Science for the Hugo Kołłątaj Agricultural University in Kraków for the year 2023.

**Institutional Review Board Statement:** Not applicable.

**Informed Consent Statement:** Not applicable.

**Data Availability Statement:** Not applicable.

**Acknowledgments:** We thank the anonymous reviewers for their constructive review, which has greatly improved this manuscript and the National Centre for Research and Development as Program Operator of the Program 'Applied Research' implemented under the European Economic Area Financial Mechanism (EEA FM) 2014–2021 and the Norwegian Financial Mechanism (NMF) 2014–2021, Scheme: Support for Ukrainian Researchers under Bilateral Fund.

**Conflicts of Interest:** The authors declare no conflict of interest.

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
