# Peer review of "European Green Deal: Study of the Combined Agricultural Aggregate"

_sustainability, doi:10.3390/su151612656_

Round 1
Reviewer 1 Report
The manuscript (sustainability-2517625) reveals that harvesting units with an asymmetrically attached header lack stability in horizontal movement. Using mathematical models and field studies, it is recommended to use self-propelled vehicles with rear-steering wheels or reversible wheeled tractors for improved stability and efficiency in crop harvesting.
The manuscript is interesting, well-written, and relevant to the field, as well as fitting within the Scopus to Sustainability journal. I get the impression that it has been resubmitted with a series of modifications.
Comments:
the abstract needs to be written in only one paragraph.
Keywords in alphabetic order;
References need to be added to lines 104-119.
All abbreviations that appear for the first time need to be described.
Figure 1 needs to have the abbreviation for vector added.
Please check the references in the Materials and Methods section.
What other methods and calculations, based on other authors and works, could be applied and compared in the modelling? For example, why is your model more robust than previous works?
Consider rewriting the conclusion based on your findings, and aim to make it clearer rather than presenting it in bullet points.
Check old references and improve;
Minor grammar and spelling.
Author Response
Thank you for your positive evaluation of our manuscript and comments!
the abstract needs to be written in only one paragraph.
This has been fixed.
Keywords in alphabetic order;
This has been fixed.
References need to be added to lines 104-119.
This has been fixed. Reference [22] is added.
All abbreviations that appear for the first time need to be described.
This has been fixed.
Figure 1 needs to have the abbreviation for vector added.
This has been fixed.
Please check the references in the Materials and Methods section.
This has been fixed.
What other methods and calculations, based on other authors and works, could be applied and compared in the modelling? For example, why is your model more robust than previous works?
There are no examples of such dynamic schemes' analysis. That is why the article authors cannot give a COMPARATIVE analysis. The reliability of our modelling can and should be assessed through the logic of the results obtained. We are ready to provide a detailed answer to any of our mathematical modelling illogical results.
Consider rewriting the conclusion based on your findings, and aim to make it clearer rather than presenting it in bullet points.
The authors believe that conclusions 3-6 of the article fully meet its aims. In our opinion, the presence of conclusions 1, 2 and 7 allows the reader better to understand the practical significance of conclusions 3-6.
Check old references and improve;
This has been fixed.
Reviewer 2 Report
I have read the European Green Deal: Study of the Stability of the Movement of the Combined Agricultural Aggregate send to Sustainability. The article is interesting. The subject is very interesting. I have some suggestions to consider before final publication. In the introduction, the Authors did not specify the purpose of the research. What is the utilitarian aspect of research.
Author Response
Thank you for your positive evaluation of our manuscript and comments!
Reviewer 2
I have read the European Green Deal: Study of the Stability of the Movement of the Combined Agricultural Aggregate send to Sustainability. The article is interesting. The subject is very interesting. I have some suggestions to consider before final publication. In the introduction, the Authors did not specify the purpose of the research. What is the utilitarian aspect of research.
According to the authors, this article's purpose is quite clearly set out in lines 128-131. The degree of achievement of the set purpose is set out in conclusions 1-6.
Reviewer 3 Report
I have gone through the review paper. The information presented by the authors is of great concern in related to "European Green Deal". The author focuses on studying the machines and harvesting methods for front harvesting and rear tillage. The Authors compared two different forms of machine configuration.
The paper is well organized and described. This work is interesting and may be accepted for publication on sustainability. Some improvements and more explanations on the work done are needed:
1. The title of the paper is too broad and not clear enough. Suggest the author to carefully consider and make revisions.
2. The combine harvester with front cutting stems and leaves and rear digging crops is similar to the research topic in this article. This type of machine is more common in potato and sugar beet harvests in European countries such as Germany and the Netherlands. It is recommended that the author cite relevant literature in the introduction and provide comparative explanations.
3. The English language should be edited by a native speaker. Minor improvements are needed.
The paper has been presented very well. Hence, I recommend this manuscript after thorough check of grammartical and typological error.
Author Response
Thank you for your positive evaluation of our manuscript and comments!
Reviewer 3
The paper is well organized and described. This work is interesting and may be accepted for publication on sustainability. Some improvements and more explanations on the work done are needed:
1. The title of the paper is too broad and not clear enough. Suggest the author to carefully consider and make revisions.
Changed
- The combine harvester with front cutting stems and leaves and rear digging crops is similar to the research topic in this article. This type of machine is more common in potato and sugar beet harvests in European countries such as Germany and the Netherlands. It is recommended that the author cite relevant literature in the introduction and provide comparative explanations.
New references with the corresponding comments of the authors have been added to the article (see Lines 111-116).
- The English language should be edited by a native speaker. Minor improvements are needed.
The paper has been presented very well. Hence, I recommend this manuscript after thorough check of grammartical and typological error.
This has been fixed.